# Cell-Autonomous and Non-Cell-Autonomous Mechanisms Concomitantly Regulate the Early Developmental Pattern in the Kelp *Saccharina latissima* Embryo

**DOI:** 10.3390/plants13101341

**Published:** 2024-05-13

**Authors:** Samuel Boscq, Bernard Billoud, Bénédicte Charrier

**Affiliations:** Morphogenesis of Macroalgae, Laboratory of Integrative Biology of Marine Models, UMR8227, Station Biologique de Roscoff, CNRS-Sorbonne University, Place Georges Teissier, 29680 Roscoff, France; samuel.boscq@sb-roscoff.fr (S.B.); bernard.billoud@sorbonne-universite.fr (B.B.)

**Keywords:** brown algae, embryogenesis, kelp, regulative, deterministic, laser ablation, image analysis, growth pattern, cell division orientation

## Abstract

Brown algae are multicellular organisms that have evolved independently from plants and animals. Knowledge of the mechanisms involved in their embryogenesis is available only for the *Fucus*, *Dictyota*, and *Ectocarpus*, which are brown algae belonging to three different orders. Here, we address the control of cell growth and cell division orientation in the embryo of *Saccharina latissima*, a brown alga belonging to the order Laminariales, which grows as a stack of cells through transverse cell divisions until growth is initiated along the perpendicular axis. Using laser ablation, we show that apical and basal cells have different functions in the embryogenesis of this alga, with the apical cell being involved mainly in growth and basal cells controlling the orientation of cell division by inhibiting longitudinal cell division and thereby the widening of the embryo. These functions were observed in the very early development before the embryo reached the 8-cell stage. In addition, the growth of the apical and basal regions appears to be cell-autonomous, because there was no compensation for the loss of a significant part of the embryo upon laser ablation, resulting in smaller and less elongated embryos compared with intact embryos. In contrast, the orientation of cell division in the apical region of the embryo appears to be controlled by the basal cell only, which suggests a polarised, non-cell-autonomous mechanism. Altogether, our results shed light on the early mechanisms of growth rate and growth orientation at the onset of the embryogenesis of *Saccharina*, in which non-cell-specific cell-autonomous and cell-specific non-cell-autonomous processes are involved. This complex control differs from the mechanisms described in the other brown algal embryos, in which the establishment of embryo polarity depends on environmental cues.

## 1. Introduction

In contrast to animals and land plants, knowledge of the mechanisms controlling embryogenesis is limited in brown algae. Many brown algae develop from isogamous fertilisation [1], suggesting autonomous development devoid of parental influence. In anisogamous or oogamous species, the formation of a polarised axis in the embryo depends on the cytoplasmic inheritance of maternal transcripts (e.g., in the brown alga *Dictyota* [2]) or male centrioles (e.g., in the brown alga *Laminaria* [3]). In these species, different molecular pathways have been shown to participate in the establishment of the first growth axis. In Fucales, the establishment of polarity and axes relies on a wide variety of factors, including light perception [4,5], gravity [6], Ca^2+^ gradients [7], ROS signalling [8], tyrosine phosphorylation [9], and cell wall determinants [10]. Processes similar to those in Fucales have been shown in Dictyotales, such as a response to light direction and the accumulation of specific compounds in the cell wall at the rhizoid pole [11]. However, the dynamics of the establishment of the first body axis differ morphologically between these two algae: the *Fucus* zygote establishes polarity through the germination of one rhizoid about 10 h after fertilisation [12,13], whereas the *Dictyota* zygote elongates 90 s after fertilisation [2,11]. In another brown alga, *Laminaria angustata* (Laminariales), the spherical egg becomes polarised just after fertilisation but before karyogamy [3]: Vacuoles accumulate in the apical region and the chloroplasts near the nucleus in the basal region (which is close to the maternal stalk). Strikingly, as the zygote elongates after karyogamy [14], its intracellular organisation becomes homogeneous, with the chloroplasts accumulating around the centred nucleus and in the periphery of the zygotic cell, where vacuoles are also scattered homogeneously [3]. This radial intracellular symmetry along the longitudinal axis of the zygote (as defined by the stalk corresponding to the remnant cell wall of the oogonium) persists until the first cell division. Despite this organisation, the first cell division—which is perpendicular to this axis—is unequal in most cases, making the basal cell slightly smaller than the apical one [3,15,16]. Later in development, the apicobasal polarisation persists: Notably, ~200-celled embryos display a narrower basal region with emerging rhizoids and a wider apical region, giving them an upside-down pear shape. At the ~1000-cell stage, the apicobasal polarisation is enhanced, with cell differentiation taking place first only in the basal region, thereby forming the haptera and the holdfast, which are attachment organs. Therefore, in this alga, the longitudinal polarisation is maintained during embryogenesis up to the adult stage. However, it is not known how this polarisation is established at the onset of embryogenesis.

Here, we studied the role of the basal and apical cells in the establishment of the apicobasal axis of the embryo in another Laminariales species: *Saccharina latissima*. More specifically, we investigated the extent to which these cells, taken separately, contribute to establishing and maintaining (1) the longitudinal axis and (2) the apicobasal polarity along this axis. We used laser ablation to specifically kill basal or apical cells at different embryo developmental stages. This technique has been developed in a wide range of organisms such as metazoans [17,18,19,20], plants [21,22], and brown algae [10,23] and, more recently, in *Saccharina latissima* [24] to identify the role of the cell neighbourhood on cell fate. We then monitored the impact of this treatment on early embryogenesis. We compared apical and basal cells at different embryogenesis steps.

## 2. Results

To study the respective roles of apical and basal cells in the early development of *Saccharina* embryos, apical or basal cells of *Saccharina* embryos were laser-ablated using a pulse-UV laser (method developed in [24]) at the 2-cell, 4-cell, and 8-cell stages, and growth was monitored every day for up to 9 days following ablation (Figure 1). Bright-field images of embryos were segmented manually up to 6 days after ablation, together with intact embryos at the same developmental stages used as controls (Figure 2 and Section 4). From these segmented images, we quantified several morphological traits to assess the role of the positional information on the growth rate and the polarity of the embryo during its development (Appendix A). To do so, we considered a snapshot of the effect 6 days after ablation and the changes in the trait over 6 days. The latter is more subtle because it depends on all the time points and hence reflects the progression of embryo growth over the 6 days. The results of statistical analyses are available in Appendix A.

### 2.1. Ablation of the Apical Cell Is Not Offset by Faster Growth

#### 2.1.1. Surface Area

To assess the requirement of the apical region and the basal region in the growth of the embryo, we laser-ablated each separately and assessed the impact on embryo growth (Figure 3). Six days after ablation, the embryos displayed a reduced surface area compared with the controls (intact embryos) (Figure 3A). Removing the apical cell at the 2-cell stage resulted in a much smaller embryo (median = 527 µm^2^) than the intact embryos at similar stages (median = 1402 µm^2^), whereas the ablation of the basal cell at the same stage led to embryos with sizes similar (median = 1240 µm^2^) to the intact embryos. The kinetics of growth was unchanged compared with intact embryos (Figure 3B, AE2 and BE2). Therefore, 2-celled embryos, once deprived of their apical cell, did not modify their growth rate to compensate for the loss of the missing cell. At the 4-cell stage, apical cell ablation resulted in embryos half as small as intact embryos after 6 days (871 µm^2^ vs. 2125 µm^2^; Figure 3A) and showed a significantly reduced growth rate (Figure 3B, AE4). At the 8-cell stage, apical cell ablation again resulted in embryos that were almost twice smaller (40% smaller: 2190 µm^2^ vs. 3676 µm^2^; Figure 3A). However, their difference to the intact embryos was not statistically significant.

The removal of the basal cell generally reduced surface area only moderately, except at the 8-cell stage, which showed significantly smaller blade surfaces (Figure 3A). In addition, the growth curves showed a reduced growth rate for ablation at the four- and 8-cell stages (Figure 3B, BE4, and BE8). This difference is due to the fact that ablation in the basal region of the embryo often induces the growth of a rhizoid from the sub-basal cell. Due to its specificity, we did not consider this cell in the segmentation, which artefactually resulted in embryos smaller than the intact embryo.

Therefore, apical and basal cell ablation during the initial embryogenesis stages (2- and 4-cell stages) appears to be detrimental to the growth of the embryo because, at these stages, the laser-ablated region of the embryo corresponds to a large proportion (25 to 50%) of the embryo, and the growth rate does not compensate for the “missing” embryo cell. We did not monitor the embryo for longer than 6 days after ablation, and therefore, we cannot exclude the possibility that compensation occurs at later stages. Ablation had more striking consequences for the apical cell than the basal cell.

#### 2.1.2. Number of Cells

From the segmented images of growing ablated embryos, the number of cells was plotted 6 days after ablation (Figure 4). Embryos in which the apical cell was ablated at the 2-cell and 4-cell stages showed a significantly lower number of cells than the control embryos (Figure 4A). This result corroborates the smaller embryo surface area observed in the same conditions (Figure 3), correlating it with a lower number of cells. The growth curve expressed in the number of cells over time shows that ablation at the 2-cell stage did not affect the rate of cell division compared with the control, but at the 4-cell stage, cells divided at a significantly lower rate (Figure 4B). This lower rate of cell division contributes to reducing the number of cells in the remaining embryo as a whole, and it further exacerbates the excision of one cell and its progeny due to laser ablation. In contrast, removing the apical cell at the 8-cell stage had no effect on cell number (Figure 4A,B, AE8).

In embryos in which the basal cells had been excised, the cell division rate was similar over time to that of the control, except for ablation at the 8-cell stage (Figure 4B, BE8 vs. CT8), resulting in the formation of embryos with fewer cells (Figure 4A). Similar to embryo surface area, this finding is an artefact due to removing the sub-basal cell from the analysis when it differentiates into a rhizoid.

Therefore, after the ablation of the apical cell or, to a lesser extent, the basal cell, the remaining portion of the embryo continued to grow at the same or slower paces as in the control embryos. This unchanged or reduced growth, along with the absence of half or one-quarter of the embryo for ablations at the 2- and 4-cell stages can account for the reduction in the size of laser-ablated embryos.

To confirm the relationship between the smaller embryo surface area and the lower number of cells, we tested whether the lower number of cells was offset by larger cell sizes. The quantification of cell surface area after image segmentation showed that cell size did not vary by more than 42% over the three developmental stages at which ablation was carried out (Appendix A). In addition, during the growth of the embryos, the cell surface area did not differ significantly from the control embryos (Appendix A): At the 2-cell stage, cells were about 192.1 µm^2^ in surface area (median for AE2; see special note in the legend of Appendix A on the initial cell size in BE2 embryos that were twice as large), and reached about 95 µm^2^ (median) after transverse and then longitudinal divisions in all embryos (calculated from Appendix A; longitudinal cell division is defined here as parallel to the growth axis of the embryo). Similar results were obtained in control embryos. Therefore, we concluded that the slight differences in cell size cannot account for the significant reduction in the size of laser-ablated embryos.

Altogether, these results show that *Saccharina* embryos, in which either the apical or, to a lesser extent, the basal cell were excised, were not able to offset the lack of the missing cell for at least 6 days after ablation. On the contrary, the remaining tissue sometimes grew more slowly. Due to ablation and lower growth curves, ablated embryos were smaller than the controls.

### 2.2. Ablated Embryos Cannot Recover the Loss of Their Initial Longitudinal Axis

We quantified the shape of the growing ablated embryos by calculating the length-to-width ratio (LWR) of the embryos every day for 6 days after ablation at the 2-cell, 4-cell and 8-cell stages. Embryo ablation resulted in the loss of the integrity of the initial embryo, which impacted de facto the shape of the embryo. Younger embryos, which lost a greater proportion of their body volume (i.e., half the initial lamina for ablation at the 2-cell stage), showed less elongated shapes. When observed 6 days after ablation at the 2-cell or 4-cell stages, the embryos displayed a lower LWR than the intact embryos (Figure 5A). They were significantly rounder than intact embryos (LWR geometrical mean = 3.48 ± 1.20 in control 2-cell stage embryos, 4.33 ± 1.24 in control 4-cell stage embryos) when the basal cell (LWR geo. mean = 2.17 ± 1.49 at the 2-cell stage, 2.65 ± 1.25 at the 4-cell stage) or the apical cell—albeit to a lesser extent (LWR geo. Mean = 2.37 ± 1.35 and 2.97 ± 1.23 respectively) –, were ablated.

The analysis of the dynamics of embryo shape showed that 2-celled embryos in which the basal cell had been ablated (BE2) did not lengthen at all for 6 days. The change in LWR as a function of time was virtually nil, and the ratio remained at 2.17, in contrast to 2-celled embryos in which the apical cell had been ablated (AE2), which continued to elongate at a rate similar to the control at the same stage (CT2) (Figure 5B). When ablation was carried out at the 4-cell stage, the removal of the apical or basal cell significantly prevented the embryo from attaining its normal shape (Figure 5A), and the analysis of the kinetics over 6 days showed that the embryo elongation rate was slower than for the control embryos (Figure 5B). When ablation took place at the 8-cell stage, the embryos developed the same anisotropic shape as the control (Figure 5A,B). Therefore, ablation affects embryo shape only at the very early stages of embryogenesis corresponding to the first two rounds of transverse cell divisions.

Overall, the ablation of the apical or basal cells at the 2- and 4-cell stages decreased the level of anisotropy in the shape of the growing embryos compared with the intact embryos. In embryos with an ablated basal cell, a more isotropic embryo shape was maintained over time (BE2 and BE4 in Figure 5B). In contrast, embryos with ablated apical cells slowly recovered a relatively more anisotropic shape (AE2, and AE4 in Figure 5B).

This impairment in shape anisotropy can in theory be due to an impairment of growth either in length or in width. Plotting the length and the width of the ablated embryos after 6 days showed that the growth in length was significantly reduced, but the growth in width showed very little change (Appendix A). This result also confirmed that the apical cell, to a greater extent than the basal cell, controls growth (Appendix A). At the 4-cell stage, both cells responded the same way, with ablated growing embryos being significantly smaller than the control. When the basal cell was ablated at the 8-cell stage, the width of the grown embryos was significantly smaller than the control, but the reason is not clear.

Therefore, the ablation of either distal cell did not allow the embryo to adjust the orientation of its growth to reshape its overall body and become as elongated as the control embryos. The effect was stronger when the basal cell was ablated at the 2- and 4-cell stages, in comparison with the apical cell.

Can cell shape be modified such that it prevents the embryo from resuming its normal shape? A detailed analysis of individual cell shape (expressed as cell LWR) showed that the shape and size of the cells of the ablated embryo were not significantly affected by ablation during 6 days after ablation (Appendix A). All cells had a similar shape with a LWR of ~1.5. When apical ablation took place at the 2-cell stage (AE2), cells from 6-day post-ablation embryos were significantly more elongated (Appendix A; LWR geometrical mean = 1.76 ± 1.32). This result is due to the fact that in this ablation context (2-cell stage and apical cell), the growth rate of the remaining basal cell was low, and cells kept dividing transversally. These factors promoted the formation of cells with an LWR increasing over time (from LWR geometrical mean = 1.12 to 1.76, Appendix A). When ablation was carried out at the 8-cell stage, ongoing longitudinal cell divisions resulted in cells with decreased LWR (from LWR geometrical mean = 2.12 to 1.53), as observed in the 8-cell control embryos (CT8, Appendix A). Altogether, except in AE2, cell shapes did not display any major changes compared with the control.

In summary, (1) embryos ablated at the 2- and 4-cell stages were not able to offset for the missing body region either in terms of biomass or in terms of morphology: they were smaller and rounder than the control embryos and embryos ablated at the 8-cell stage; (2) the roles of the apical and basal cells are different: the removal of the apical cell had the greatest impact on growth, but the basal cell had the strongest impact on embryo shape.

### 2.3. Basal Cells Negatively Control Longitudinal Cell Division

A closer examination of the cell division pattern of laser-ablated embryos showed that the emergence of the first longitudinal division was affected. For all stages and both ablation regions, ablated embryos displayed the earlier emergence of longitudinal divisions compared with the intact embryos, in which the first longitudinal division routinely occurred at the 8-cell stage (Figure 6), usually in the apical region. After apical ablation, longitudinal divisions started around the seven-cell stage regardless of the stage of ablation (AE2, AE4, and AE8). Conversely, in basally ablated embryos, the earlier the ablation was, the earlier the emergence of longitudinal cell divisions was. At the 2-cell and 4-cell stages, basal ablation resulted in the first longitudinal division as early as the three-cell (BE2) and 4-cell stages (BE4), respectively. This division occurred earlier than in intact embryos and apically ablated embryos. This variation in the timing of transversal divisions suggests that, for the orientation of cell division, the apical cell behaves non-autonomously and requires the basal cell to maintain exclusively transverse cell divisions. This influence diminishes gradually until the 8-cell stage.

## 3. Discussion

At first sight, the early development of the embryo of *Saccharina latissima* appears extremely simple. The elongated zygote first divides transversally into two cells of about the same size and then grows as a stack of eight cells through two additional rounds of transverse cell divisions for each daughter cell. Thereafter, the embryo grows as a flat monolayered lamina [16]. In the 8-cell stack and the bi-dimensional, flat lamina, the cells look similar to their cell neighbours, with no signs of specific cell differentiation. Therefore, until molecular markers are identified, this simple morphology is an asset for teasing apart the mechanisms that control growth but also a disadvantage for tracking the establishment of polarisation. Here, we took advantage of this stereotypical pattern of cell division orientation in this alga to decipher how growth per se, on the one hand, and the direction of growth, on the other hand, are controlled.

Using laser ablation, we investigated the role of the apical and basal cells during early embryogenesis by monitoring over time the growth rate and the orientation of cell division in embryos ablated in either the apical or basal regions at three early developmental stages.

### 3.1. Concomitant Cell-Autonomous and Non-Cell-Autonomous Controls of Growth

On the one hand, basal cells were unable to compensate for the loss of the ablated apical region. Six days after the ablation of the apical cell at the 2-cell stage, embryos were at least half as small as intact embryos at the same stage. Their growth curve was similar to that of intact embryos. Therefore, the ablated apical region was not replaced by a higher growth rate and, in the time window of 6 days after ablation, the basal cells acted independently of the apical cell(s), i.e., they were cell-autonomous. On the other hand, the control of the longitudinal cell divisions was non-cell-autonomous. In intact *Saccharina* embryos, longitudinal cell division is initiated around the 8-cell stage [16]. Therefore, if the initiation of longitudinal cell division was cell-autonomous, after the ablation of one cell in a 2-celled embryo, the cell progeny of the remaining cell should shift to longitudinal cell division when the embryo is close to the 4-cell stage (corresponding to the progeny of the remaining cell after two rounds of cell division). However, regardless of the region of the ablated cell, longitudinal cell division did not start at the 4-cell stage. When the apical cell was ablated, longitudinal cell division began at the seven-cell stage, i.e., very similar to intact embryos. This finding suggests that the control of longitudinal cell division in basal cells depends on the number of cells in the embryo. In contrast, when the basal cell was ablated, the embryo started dividing longitudinally immediately after ablation, independently of the stage of ablation. In both cases, the control on longitudinal cell division thus appears to be non-cell-autonomous. To account for the two responses, we hypothesise that the basal cell emits a signal inhibiting longitudinal cell divisions towards the apical region. Therefore, when the apical cell is ablated, the remaining basal cells must produce a sufficient number of cells to outdistance the source of this inhibitory signal, i.e., up to the seven-cell stage at least. When the basal cell is ablated, the inhibitory signal is cancelled, and the remaining cells immediately commit to longitudinal cell divisions. Plasmodesmata may mediate the transport of such a signal because they have been shown to be able to facilitate symplastic exchanges of elements up to 20 kDa [25] in *Saccharina* sporophytes and gametophytes [26] and in late embryos [16].

### 3.2. The Impact of Ablation Is Greatest before the Four-Cell Stage

We have shown that, in the embryo of *Saccharina latissima*, the ablation of the apical or basal cells only affects the very beginning of embryogenesis, i.e., when the embryo reaches up to four cells. After that stage, the removal of basal or apical cells no longer significantly impacts the embryogenetic developmental pattern or growth. The immediate explanation is that the removal of one cell at the 8-cell stage corresponds to a proportionally smaller area of the embryo, compared with 2- and 4-celled embryos in which half and one-quarter of the embryo are removed, respectively. Alternatively, although cell growth is cell-autonomous just after fertilisation, the growth of the 8-cell-stage embryos may be subject to a regulative regime in which cell fate globally depends on the rest of the embryo and locally on the fate of neighbouring cells. Such non-deterministic development at later embryogenetic stages has already been observed in brown algae *Fucus* [10,23] and *Ectocarpus* [27], in plants [28], and in animals [29]. The 8-cell stage corresponds to a major transition in the embryogenetic pattern of *Saccharina* [16], which may coincide with a change in the allocation of cell function. In early embryogenesis, each cell seems to have its own identity, with cells of embryos of more than eight cells (corresponding to Phase II [16]), all behaving the same way in terms of the orientation of cell division, growth rate, and lack of cell differentiation. Thus, in this scenario, the destruction of apical or basal cells in 8-celled embryos has no more impact than the destruction of any other cell of these embryos, i.e., nearly no impact. Testing this hypothesis requires irradiating the cells located between the apical and basal cells, which we did not do.

In light of our results, we propose an overall model of early embryogenesis in *Saccharina latissima.* Before the 8-cell stage, the basal cell inhibits longitudinal cell divisions in the apical cell or apical region of the embryo (Figure 7). The basal and apical regions grow through cell division, but the apical region grows more actively than the basal cell (Figure 7, green arrowhead indicating the growth gradient). Drawing on our observation that neither the basal cells nor the apical cells were able to compensate for the loss of a portion of the embryo after 6 days, our model considers that the growth activity of each cell is cell-autonomous. Once growth ensures that the apical-most cells are distant enough from the basal cell, the inhibition of longitudinal cell division (Figure 7; blue ‘T’ and blue arrowhead indicating the inhibition gradient) is lifted, and cells divide longitudinally. The embryo starts growing in width away from the basal cells, giving the embryo its recognisable upside-down pear shape (Figure 7). According to this model, a signal inhibiting longitudinal cell divisions diffuses from the basal cell towards the rest of the embryo. Therefore, the embryogenesis of *Saccharina* is characterised by an interplay between (1) cell-autonomous mechanisms that control growth activity and (2) a basal, positional signal that inhibits longitudinal cell division in the apical region. Therefore, *Saccharina* combines cell-autonomous and non-cell-autonomous controls of its embryogenesis, with the control of the growth rate being cell-autonomous and the control of growth orientation being non-cell-autonomous. A recent study showed that the dead cell that connects the embryo to the maternal gametophyte is necessary for the establishment of the apicobasal axis in the *S. latissima* embryo [30]. It is therefore likely that the non-cell-autonomous mechanism that confers to the basal cell its inhibitory effect on longitudinal cell division is linked to a signal transmitted by the maternal gametophyte.

The longitudinal cell divisions spread to the middle of the embryo, allowing the embryo to develop an upside-down pear-shaped lamina. Note that the positions of the longitudinal cell division are not systematically in the upper-most part of the embryo. However, it never occurs in the basal cell (personal observations).

Similar studies have been carried out regarding other brown algae. In *Fucus spiralis*, the first cell division of the zygote is unequal and asymmetrical. The large apical cell gives rise to the thallus, and the small basal cell gives rise to the rhizoid. Ablation of either rhizoid or thallus cells shows that the cell wall dictates the fate of the newly growing cells [31]. A rhizoid or thallus cell devoid of its cell wall (i.e., protoplast) resumes the initial zygotic program from the beginning, but cells isolated from the rest of the embryo with their own cell wall grow and maintain their original cell identity. However, when an isolated cell surrounded by its cell wall grows in contact with the wall of the ablated neighbouring cell, it adopts the identity of the neighbouring cell, including its cell division pattern [10]. Therefore, a thallus cell growing in contact with a rhizoid cell wall initiates the growth of a polarised rhizoid through transverse cell divisions, but a rhizoid cell in contact with the thallus cell wall stops tip growth and divides longitudinally, producing a spherical group of cells characteristic of the thallus. Thus, the cell wall seems to contain cell fate determinants that can modify the initial cell fate. In both cases, the cells respond to local determinants present either in their own cell wall or in that of their neighbours when they come into contact with it. Later in the development of *Fucus*, different results are observed. Ablation of rhizoid cells at different positions in embryos having more than eight cells leads to a gradient in cell response: Increasing distance from the rhizoid pole or the presence of fewer rhizoid cells increases the likelihood that rhizoids emerge from the base of the thallus cell [23]. Therefore, an inhibitory signal synthesised by the rhizoid cells appears to be transported across several cells in the embryo, where it inhibits the formation of additional, adventitious rhizoids. Hence, cell-autonomous and non-cell-autonomous mechanisms come into play, with a predominant role of the cell wall providing local positional information. However, in contrast to *Saccharina*, these mechanisms occur successively and are not synchronised in the early stages.

In *Ectocarpus*, which belongs to the order Ectocarpales, an order distinct from *Saccharina* (Laminariales) and *Fucus* (Fucales), axial and polarised growth are established at the onset of embryogenesis. As in *Fucus*, the spherical zygote germinates into an elongated, apically growing cell, which forms a filament after several transverse cell divisions. With time, the elongated cells differentiate into round cells. In response to the ablation of the apical elongated cells, lateral branching is induced, and round cells start filling in for the missing region by growing elongated cells [27]. This finding, obtained in embryos of about 10 cells, shows that the development of *Ectocarpus* is non-deterministic. However, it is not known whether cells behave more cell-autonomously at earlier stages, because cells of the two cell types were not isolated from filaments with fewer than ten cells. Nonetheless, monitoring the establishment of elongated vs. round cells from zygote germination to the 10-cell stage led to a model of a mechanism by which cells differentiate according to the shape of their adjacent neighbours, involving local positional information [32]. Hence, in *Ectocarpus*, as in *Fucus*, embryo cells appear to first respond to very local positional information to establish the initial growth axis, before a more global mechanism takes over at the level of the organism, possibly involving the diffusion of signals through plasmodesmata.

Therefore, in brown algae, natural selection has resulted in several different strategies to control the formation of growth axes.

## 4. Materials and Methods

### 4.1. Algal Culture

The culture and production of *Saccharina latissima* (Phaeophyceae, Laminariales, Arthrothamnaceae) embryos was carried out as in [33,34]. Fixed female and male genotypes were used by selecting one female gametophyte (F_1_) and one male gametophyte (M_1_) from the same sporophyte collected from the beach of Perharidy in Roscoff, northwestern France (48°43′33.5″ N 4°00′16.7″ W). When crossed, F1 and M1 presented very stable and effectively growing embryos. Embryos and gametophytes were all cultivated in Provasoli-enriched natural seawater. Gametophytes (F_1_ and M_1_) were cultivated together and vegetatively propagated as described in [34]. Gametes were obtained from the maturation of gametophytes under 16 μmol photons·m^−2^·s^−1^ white light intensity and 14:10 light–dark photoperiod at 13 °C. If necessary, they were kept under such conditions to grow to the required stage.

### 4.2. Laser Irradiation of Embryo Cells at Different Developmental Stage

Developing embryos growing in glass-bottom Petri dishes were selected under a flow hood using an inverted microscope (LSM 880 Zeiss confocal microscope, Carl Zeiss S.A.S., Rueil-Malmaison, France). The laser ablation of either the apical or basal cell was carried out on embryos as described in [24]. The perforation of the cell wall and plasma membrane resulted in the emptying of most of the intracellular contents in response to decreased turgor pressure. Monitoring the growing embryos daily confirmed that the irradiated cells were dead (Figure 2A). Intact, non-targeted embryos in the same dish were used as controls. Laser irradiation was performed on embryos at the 2-cell (E2), 4-cell (E4), and 8-cell (E8) stages. To avoid any stress during transportation to the cytology platform (MRiC, Rennes, France), embryos were transported in a cooler set to 13 °C. Irradiated embryos, along with control (non-irradiated) embryos were monitored using time-lapse microscopy in a bright field (Figure 1) for up to 9 days.

### 4.3. Image Acquisition

All time-lapse microscopy experiments of growing embryos were recorded under a bright field or through epifluorescence with a Leica Inverted phase contrast or Leica DMI8 microscope (Leica Microsystèmes SAS, Nanterre, France) equipped with a DFC450C camera (Leica Microsystèmes SAS, Nanterre, France) with acquisition intervals of 24 h for a duration of at least 9 days after laser irradiation. The required temperature and light were set in a carbonate-glass chamber fitted to the microscope and equipped with a thermostatically controlled airflow system (Cube and Box, Life Imaging Services, GmbH, Basel, Switzerland) and commercially available LED white-light sources.

### 4.4. Manual Segmentation

Automatic segmentation proved to be challenging due to the constant pigmentation changes in the cells transitioning from high to low colouring. Additionally, daily exposure to UV light used to reveal the cell wall staining chemical Calcofluor White proved to be detrimental to algal survival, rendering it unsuitable for precise acquisition of the cell wall outlines. Therefore, segmentation was performed manually from images obtained from bright-field microscopy. Images from the time lapse with intervals of 24 h were segmented using Fiji image processing software (ImageJ2 version 2.9.0). We selectively segmented predominantly flat-growing embryos. However, when embryos grew with a slight inclination relative to the surface of the Petri dish, Z-stacks were acquired, and segmentation was carried out on the projection of the Z-stacks using the Stackfocuser plugin (https://imagej.net/ij/plugins/stack-focuser.html (accessed on 13 April 2024)) to improve the accuracy of the segmentation. The outlines were then implemented using the vector graphics editor Inkscape (v1.2.2, https://inkscape.org (accessed on 13 April 2024)). Drawings were saved as scalable vector graphics (svg) files. The cell contours of all Z-stacks from one given time series were compared, from which adjustments were carried out when necessary.

### 4.5. Quantitative Morphometry

These files were processed by dedicated software written in object-oriented Python 3 [35] that we developed called blade_painter. Reading the svg file, blade_painter extracts various geometric properties for cells and blades [36], namely (a) each cell contour, from which the surface area was directly derived; (b) convex hull, used to compute the minimal bounding rectangle (MBR), whose axes were used to measure the cell length-to-width ratio; (c) the blade area, computed as the sum of cell areas; and (d) the orientation of the main and secondary axes of the blade (length and width), computed according to [37]. When applicable, measures were log-transformed, and then the increase rate was estimated by fitting a line to the observed points and extracting the slope of this line.

### 4.6. Statistical Tests

Quantitative data were analysed using pairwise Mann–Whitney U-tests, either one-tailed in cases for which the ablated cell was expected to affect the studied parameter in a given direction, or, by default, two-tailed. A Student *t*-test was used for the analysis of the cell LWR. Statistics were carried out on the data processing libraries available under Python: numpy, scipy, and pandas; *p*-values lower than α = 5 × 10^−2^ were considered significant.

## 5. Conclusions

Brown algae, like animals, have different ways of regulating the establishment of their body axes. Embryogenetic mechanisms are classically termed deterministic (formerly “mosaic”) or non-deterministic (formerly “regulative”) based on the inheritance of maternal polarity cues via the egg vs. the perception by the embryo cells of position-dependent cues from the environment (including neighbouring cells) [38,39]. *Saccharina* uses both simultaneously to form its early and characteristic upside-down pear-shaped embryo, a morphological shape that lasts until the adult stage.

## Figures and Tables

**Figure 1 plants-13-01341-f001:**
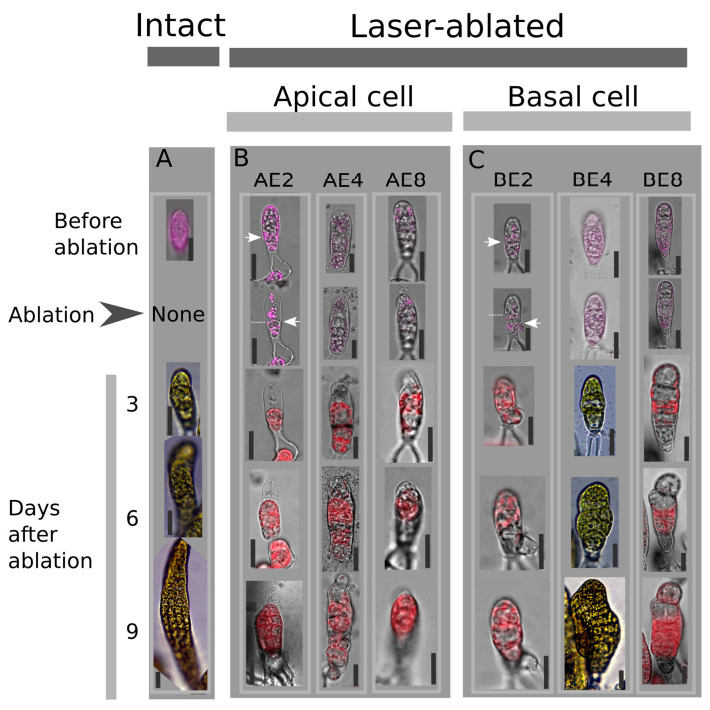
Ablation of apical and basal cells in *Saccharina latissima* embryos and time-dependent monitoring of embryo development. Intact embryos not targeted by laser ablation are shown as controls (**A**). Morphologies of developing embryos after ablation of the apical or basal cell are shown before ablation and up to 9 days after ablation. Ablation of the apical cell took place at the 2-cell stage (AE2), 4-cell stage (AE4), and 8-cell stage (AE8) (**B**). Similarly, ablation of the basal cell took place at the 2-cell stage (BE2), 4-cell stage (BE4), and 8-cell stage (BE8) (**C**). To visually confirm ablation, we monitored the extrusion of chloroplasts during ablation (autofluorescence in red or pink; Exc/Em: 570/680 nm) (see photos in the row marked “ablation”). White arrows in the AE2 and BE2 columns display the first transverse cell wall before and after ablation (the dotted line shows the position of the transverse cell wall before ablation). In AE2, the transverse cell wall did not change position after ablation but shifted downwards in BE2. A representative image of the observed morphology is shown. Monitoring was performed for three independent experiments, cumulating n = 15 (control), 9 (AE2), 11 (AE4), 4 (AE8), 10 (BE2), 18 (BE4), and 10 (BE8). Scale bars represent 20 µm.

**Figure 2 plants-13-01341-f002:**
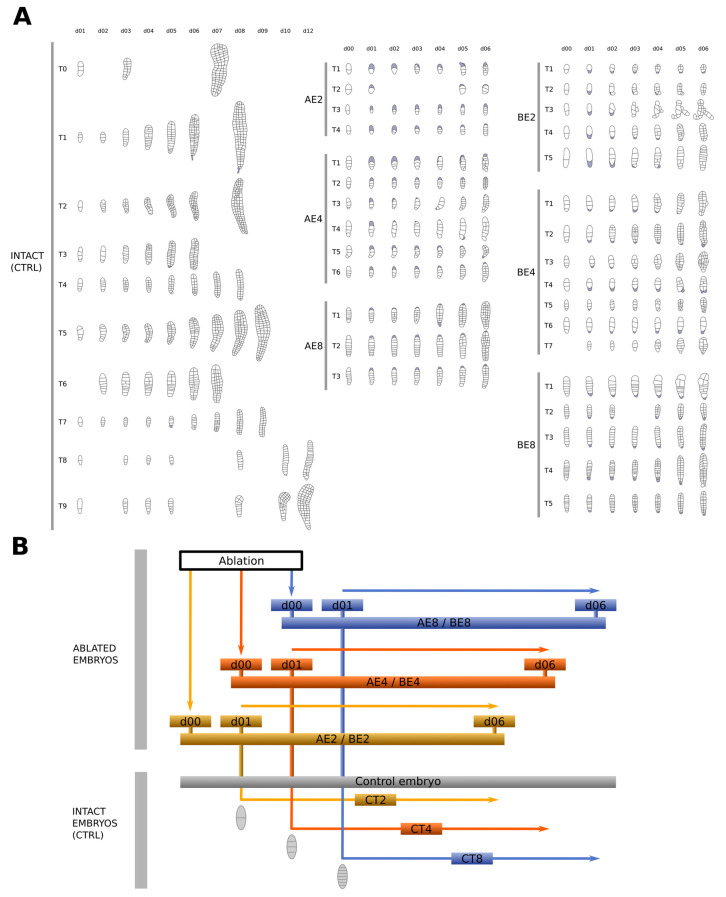
Segmentation of growing laser-ablated *Saccharina* embryos and comparison with intact control embryos: (**A**) Image segmentation was performed manually for each time point of the growth kinetics (one per day). Several replicates (Tn) of ablated embryos and intact embryos are shown: (**left** panel) control (CTRL); (**middle** panel) ablated apical cells (named A for apical); and (**right** panel) ablated basal cells (named B for basal) at the 2-cell, 4-cell, and 8-cell stages (E2, E4, and E8, respectively). All segmented embryos are shown at the same scale. Grey areas indicate ablated cells. (**B**) Chart showing how intact embryos were used to compare with ablated embryos: (**top** panel) ablation took place on day 0 (d00) at the 2-cell, 4-cell, or 8-cell stage and was monitored every day for 6 days (d06); (**bottom** panel) intact embryos were monitored from the same developmental stages (CT2, CT4, and CT8) for the same duration (6 days each). Hence, a single intact embryo can be used for comparison of several ablation stages.

**Figure 3 plants-13-01341-f003:**
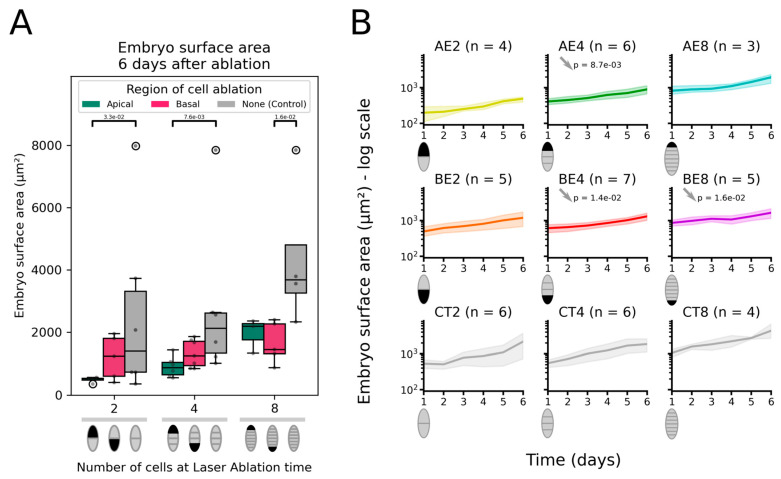
Surface area of the laser-ablated embryos. Surface area (µm^2^) of the embryo blades, after ablation at the 2-, 4-, or 8-cell stage, compared with control (intact) embryos 6 days after laser irradiation (**A**). The diagrams below the graph show which cell was ablated (black area). In the box-and-whisker plots, the horizontal middle line of each box shows the median, and the boxes delimit the first and third quartiles. Whiskers indicate the observation interval up to 1.5 times the interquartile difference. Significantly (*p*-value < 5 × 10^−2^) smaller surface area in treated embryos than in control embryos (one-tailed Mann–Whitney U-test) is shown as a bar topped by the *p*-value. (**B**) Change in surface area (shown in log scale) for embryos apically (AE) or basally (BE) ablated at the 2-, 4-, or 8-cell stage compared with control (CT) embryos starting at the same developmental stage (hence, CT2, CT4, and CT8 for control embryos at the 2-, 4- or 8-cell stages). The thick lines show the mean values, and the shaded areas show the 95% confidence intervals. The area increase rate, estimated as the slope of the regression line adjusted to the logarithm of the surface area, is compared with the control (CT at the same stage), and in cases when the difference is significant, the *p*-value of the two-tailed Mann–Whitney U-test is indicated near the grey arrow, indicating the trend in difference. See Appendix A for raw data and Appendix A for the values of the slope and *p*-values of the statistical tests.

**Figure 4 plants-13-01341-f004:**
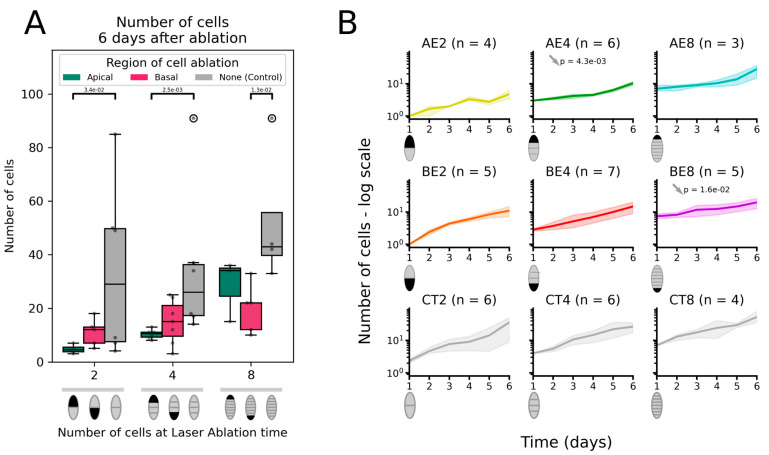
Cell number of the laser-ablated embryos. Cell number in the embryo blades after ablation at the 2-, 4-, or 8-cell stage, compared with control (intact) embryos 6 days after laser irradiation (**A**). See Figure 3A for the description of the box-and-whisker plot and diagrams below. Significantly (*p*-value < 5 × 10^−2^) lower number of cells in treated embryos than in control embryos (one-tailed Mann–Whitney U-test) are shown as a bar topped by the *p*-value. (**B**) Kinetics of the change in cell number (shown in log scale) over time (days) for embryos apically (AE) or basally (BE) ablated at the 2-, 4-, or 8-cell stage compared with the kinetics in control (CT) embryos starting at the same developmental stage (hence, CT2, CT4, and CT8 for control embryos at the 2-, 4-, or 8-cell stages). The growth rate, estimated as the slope of the regression line adjusted to the logarithm of the number of cells, is compared with the control (CT at the same stage), and in cases when the difference is significant, the *p*-value of the two-tailed Mann–Whitney U-test is indicated near the grey arrow, indicating the trend in difference. See Appendix A for raw data and Appendix A for the values of the slope and *p*-values of the statistical tests.

**Figure 5 plants-13-01341-f005:**
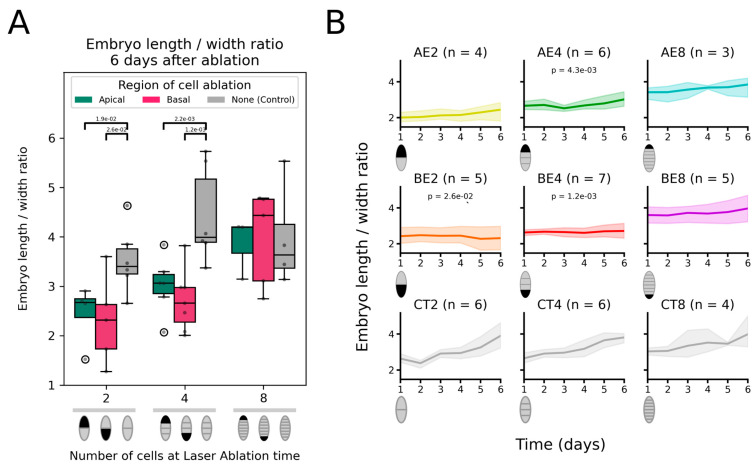
Shape of the laser-ablated embryos. Embryo shape, characterised by its blade length-to-width ratio, after ablation at the 2-, 4-, or 8-cell stage, compared with control (intact) embryos 6 days after laser irradiation (**A**). See Figure 3A for the description of the box-and-whisker plot and diagrams below. Significantly (*p*-value < 5 × 10^−2^) lower length-to-width ratio in treated embryos than in control embryos (one-tailed Mann–Whitney U-test) are shown as a bar topped by the *p*-value. (**B**) Kinetics of length-to-width ratio for embryos apically (AE) or basally (BE) ablated at the 2-, 4-, or 8-cell stage, compared with the kinetics of length-to-width ratio in control (CT) embryos starting at the same stage (hence, CT2, CT4, and CT8 for control embryos at the 2-, 4-, or 8-cell stages). See Figure 3B for a description of the line plots. The shape change rate, estimated as the slope of the regression line adjusted to the length-width ratio, is compared with the control (CT at the same stage), and in cases when the difference is significant, the *p*-value of the two-tailed Mann–Whitney U-test is indicated. See Appendix A for raw data and Appendix A for the values of the slope and the *p*-values of the statistical tests.

**Figure 6 plants-13-01341-f006:**
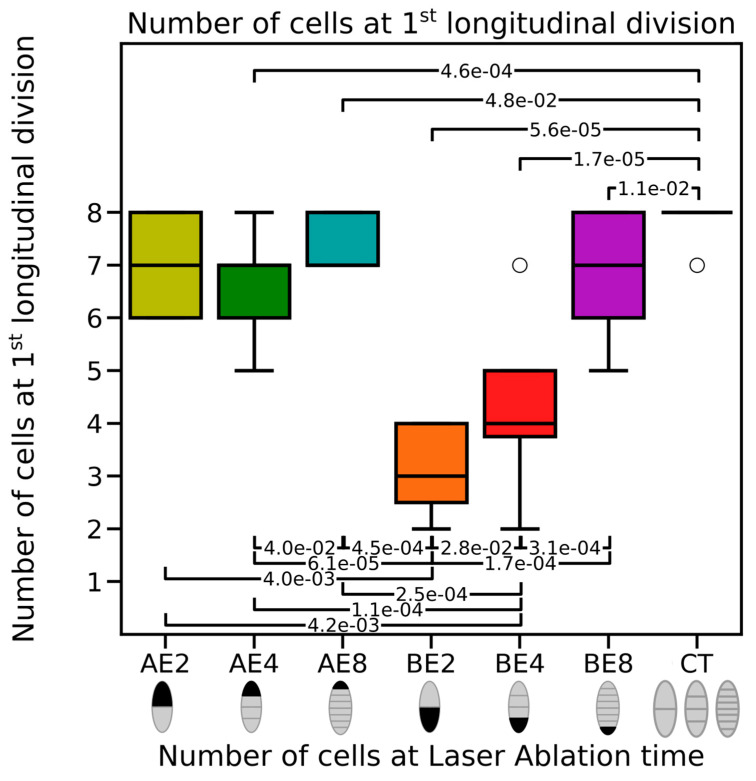
Stage of first longitudinal cell division. Number of cells at the time of the first longitudinal cell division for embryos apically (AE) or basally (BE) ablated at the 2-, 4-, or 8-cell stage compared with control (CT) embryos. For a description of the box-and-whisker plot, see Figure 3A; *p*-values are indicated for each significant (*p*-value < 5 × 10^−2^) pairwise test (two-tailed Mann–Whitney U-test with continuity correction). Sample size: n = 4 (AE2), 11 (AE4), 7 (AE8), 11 (BE2), 16 (BE4), 9 (BE8), and 10 (CTR).

**Figure 7 plants-13-01341-f007:**
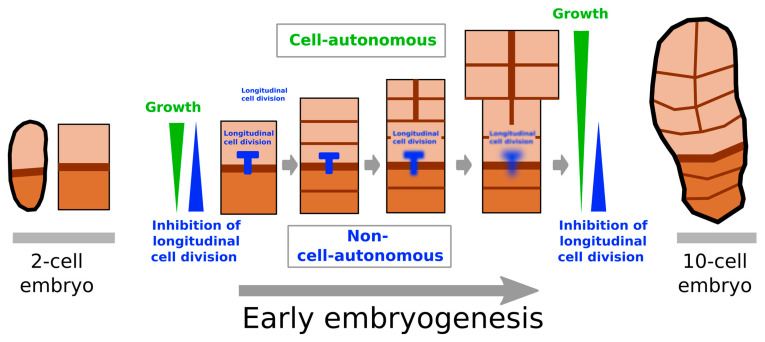
Summary of the role of the apical and the basal cells in the early embryogenesis of *Saccharina latissima.* Cuboids represent the apical (light brown) and basal (medium brown) cells of a 2-celled embryo (left, the first transverse cell wall indicated by a thick dark brown line) through to a 10-celled embryo (the initial position of the first transverse cell division is shown). Growth is more active in the apical region than in the basal region of the embryo (growth gradient shown as a green arrowhead), while inhibition of longitudinal cell division is stronger in the basal cell than in the apical cell (inhibition gradient shown as a blue arrowhead). The basal cell inhibits longitudinal cell divisions in the apical cells (blue ‘T’ overlapping the apical and the basal boundaries). The growth gradient progresses with embryo growth (growth gradient on the right), while the gradient of inhibition of longitudinal cell division remains at its initial amplitude (inhibition gradient on the right) because it depends directly on physical distance. Therefore, the apical region is now out of the range of action of the basal inhibition and starts dividing longitudinally. Moreover, the effect of the gradient diminishes during early embryo development, becoming almost nil at the 8-cell stage (blurred blue “T”).

## Data Availability

The original contributions presented in this study are included in the article/Appendix A; further inquiries can be directed to the corresponding author.

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
