# Peer review of "Cell-Autonomous and Non-Cell-Autonomous Mechanisms Concomitantly Regulate the Early Developmental Pattern in the Kelp Saccharina latissima Embryo"

_plants, 2024, doi:10.3390/plants13101341_

Round 1
Reviewer 1 Report
Comments and Suggestions for Authors
This manuscript addresses the regulation of embryogenesis of Saccharina latissima by ablating the apical or basal cell during different stages of early embryo development. The authors show that the apical cell plays a role in the elongation of the embryo as embryo’s with ablated apical cells are shorter and wider after 6 days. The basal cell appear to play a regulatory role in inhibiting longitudinal cell divisions, which is restricted to the 8 cell stage.
Unlike in Fucus and Ectocarpus, which are both broadcast spawners, Saccharina eggs remain attached to their oogonium on the female gametophyte. In a preprint on BioRxiv (Boscq et al, 2024), the authors claim that cues from the oogonium play a role in establishing the polarity of the developing embryo, whereas polarity establishment in Fucus depends fully on environmental cues. Earlier studies were not able to provide a clear answer to the question during what developmental stage the maternal signal perceived by the basal cell of the embryo is translated into polarity establishment in the embryo. This study attempts to addresses this issue.
Although the presented work does not give insight in the regulatory pathways during embryogenesis, it helps to establish the regulatory roles of the apical and basal cells. Generally, the data and analyses appear sound. However, perhaps due to the format of the manuscript, the results appear somewhat detached from the existing knowledge about the role of the oogonium/female gametophyte in establishing polarity in the embryo. Why does embryo development require an inhibitory signal for longitudinal cell division during the 8-cell stage and not during earlier stages? This observation suggests that, once 2 and 4 cell embryo’s with an ablated basal cell reach the 8 cell stage, the cell that was located proximal to the ablated basal cell must have acquired a basal cell fate. This cell, however, lacks the connection with the oogonium that is thought to be responsible for establishing the apical-basal axis in the embryo. Since the disruption of the oogonial connection occurs at all developmental stages, I would encourage the authors to share their thoughts on possible reasons for the stage specific induction of longitudinal divisions after ablation of the basal cell that they observe.
Some other points that the authors could address:
-Laser ablation is an excellent, and more frequently used tool to establish the contribution of specific cells in the control of polarization. Although it is non-invasive, killing a cell is likely to induce stress responses. It would be useful to discuss why the authors consider the observed responses as defective development rather than a response to stress. This is particularly relevant in the cases that growth reduction was observed after ablation.
-It is convincing that the authors used statistics to determine the significance of differences in growth rate. Since the graphs appear quite similar, it would be nice to include the actual p-values for each graph presented in the figures, so that readers can accurately compare the different groups. When reading the manuscript, it took me a while to understand the meaning of the stars and arrows in the figures. My confusion was mostly caused by their positioning next to the line; the arrows appear to indicate a specific part of the lines and the stars appear associated with the arrows. I would suggest to reposition the stars and arrows and to include the p-values in the figures.
Author Response
Comments and Suggestions for Authors
This manuscript addresses the regulation of embryogenesis of Saccharina latissima by ablating the apical or basal cell during different stages of early embryo development. The authors show that the apical cell plays a role in the elongation of the embryo as embryo’s with ablated apical cells are shorter and wider after 6 days. The basal cell appear to play a regulatory role in inhibiting longitudinal cell divisions, which is restricted to the 8 cell stage.
Unlike in Fucus and Ectocarpus, which are both broadcast spawners, Saccharina eggs remain attached to their oogonium on the female gametophyte. In a preprint on BioRxiv (Boscq et al, 2024), the authors claim that cues from the oogonium play a role in establishing the polarity of the developing embryo, whereas polarity establishment in Fucus depends fully on environmental cues. Earlier studies were not able to provide a clear answer to the question during what developmental stage the maternal signal perceived by the basal cell of the embryo is translated into polarity establishment in the embryo. This study attempts to addresses this issue.
Although the presented work does not give insight in the regulatory pathways during embryogenesis, it helps to establish the regulatory roles of the apical and basal cells. Generally, the data and analyses appear sound. However, perhaps due to the format of the manuscript, the results appear somewhat detached from the existing knowledge about the role of the oogonium/female gametophyte in establishing polarity in the embryo.
Answer from the authors: This work focuses on the respective roles of the apical and basal cells of the embryo proper, and not on the relationship between the embryo and the maternal gametophyte. The latter is the topic of another study and we believe that in order to determine the control of embryogenesis as a whole, it is important to separate the different levels of control, in the first place
Why does embryo development require an inhibitory signal for longitudinal cell division during the 8-cell stage and not during earlier stages?
Answer from the authors: It seems that there is a misunderstanding about our main results: the inhibition of longitudinal cell division occurs from the 2-cell stage, where its effect is the strongest, up to the 8-cell stage, when the effect is the weakest and nearly nil. This is explicitly indicated throughout the text, e.g. in the Abstract (lines 22-23): “These functions were observed in the very early development, before the embryo reaches the 8-cell stage.” In the Results section (lines 158-159): “Removing the apical cell at the 2-cell stage resulted in a much smaller embryo (median = 527 µm2) than the intact embryos at similar stages.” And in all the sub-sections where the results first describe the impact of ablation at the 2-cell stage, then at the 4-cell stage, to end with the 8-cell stage. We did not revise the text regarding this comment.
This observation suggests that, once 2 and 4 cell embryo’s with an ablated basal cell reach the 8 cell stage, the cell that was located proximal to the ablated basal cell must have acquired a basal cell fate.
Answers from the authors: First, we don’t understand the expression “ablated basal cells must have acquired a basal cell fate”. The term “must” is not appropriate. In addition, we cannot tell how ‘basal cell fate’ is defined. In early intact embryos, we define the basal cell not by its function in e.g. development or physiology, but by its position in space, relative to the maternal stalk. Therefore, when it is ablated, the sub-basal cell becomes the most basal spatially. Whether it behaves as the initial basal cell is unknown, because the basal cell has no systematic and apparent morphological function at that stage. Later in development, rhizoids emerge from the basal cell. However, our study did not monitor late development.
This cell, however, lacks the connection with the oogonium that is thought to be responsible for establishing the apical-basal axis in the embryo. Since the disruption of the oogonial connection occurs at all developmental stages, I would encourage the authors to share their thoughts on possible reasons for the stage specific induction of longitudinal divisions after ablation of the basal cell that they observe.
Answers from the authors: Again, we are not sure that we correctly understand this comment. Revised figure 7 presents the model in which the inhibition of longitudinal cell division progressively weakens from the 2-cell stage to the 8-cell stage. We can propose this model specifically because we observed the release of inhibition when the basal cell was ablated, from the 2-cell to the 8-cell stage. This is shown in Figure 5A & B. Reviewer N°1 can also consult our BioRXiv preprint which shows that the maternal stalk is also necessary for the establishment of the longitudinal growth axis (https://biorxiv.org/cgi/content/short/2024.01.07.574535v1). In the discussion of the revised version, we discuss the possibility that the basal cell conveys a signal transmitted by the maternal stalk to the apical region of the embryo. However, the nature of this signal is still unknown and therefore, this hypothesis needs further work before it can be confirmed.
Some other points that the authors could address:
-Laser ablation is an excellent, and more frequently used tool to establish the contribution of specific cells in the control of polarization. Although it is non-invasive, killing a cell is likely to induce stress responses. It would be useful to discuss why the authors consider the observed responses as defective development rather than a response to stress. This is particularly relevant in the cases that growth reduction was observed after ablation.
Answers from the authors: We thank Rev N°1 for this interesting question. We observed a gradual response to ablation, which depends on the developmental stage at which ablation is carried out. If the morphological response was due to the stress only, then this response would be similar at all developmental stages. Therefore, although we cannot exclude the possibility that the observed signal includes some stress response due to the ablation itself, it must also include a factor related to the removal of half or a quarter of the embryo.
-It is convincing that the authors used statistics to determine the significance of differences in growth rate. Since the graphs appear quite similar, it would be nice to include the actual p-values for each graph presented in the figures, so that readers can accurately compare the different groups.
Answers from the authors: In the revised version, we report actual p-values instead of using asterisks.
When reading the manuscript, it took me a while to understand the meaning of the stars and arrows in the figures. My confusion was mostly caused by their positioning next to the line; the arrows appear to indicate a specific part of the lines and the stars appear associated with the arrows. I would suggest to reposition the stars and arrows and to include the p-values in the figures.
Answers from the authors: Regarding the figures showing curves, the arrow indicates the trend: increase or decrease of the curve as a function of the developmental stage. The asterisks indicated the level of the p-value corresponding to this overall trend, and therefore, the two symbols were combined. In the revised version, we replaced the asterisks with the actual p-values. The arrow remains over the curve.
We thank Rev N°1 for their questions and hope that this revised version will improve the understanding of our manuscript.
Reviewer 2 Report
Comments and Suggestions for Authors
All my detailed comments are in PDF file and in review which is sent to Editor.

Author Response
Answers from the authors: Our answers are incorporated in the PDF files with the questions of Rev N°2.
We thank Rev N°2 for their questions and hope that this revised version will improve the understanding of our manuscript.
